# Denosumab Attenuates Glucolipotoxicity-Induced β-Cell Dysfunction and Apoptosis by Attenuating RANK/RANKL Signals

**DOI:** 10.3390/ijms241210289

**Published:** 2023-06-17

**Authors:** Sheng-Chieh Lin, Sing-Hua Tsou, Chien-Yin Kuo, Wei-Liang Chen, Kuan-Wen Wu, Chih-Li Lin, Chien-Ning Huang

**Affiliations:** 1Institute of Medicine, Chung Shan Medical University, Taichung 402, Taiwan; phoenix33343@gmail.com (S.-C.L.); a9704119@hotmail.com (C.-Y.K.); 2Department of Orthopaedics, Chung Shan Medical University Hospital, Taichung 402, Taiwan; 3Department of Medical Research, Chung Shan Medical University Hospital, Taichung 402, Taiwan; zinminid@gmail.com; 4Department of Surgery, Chung Shan Medical University Hospital, Taichung 402, Taiwan; 5Department of Internal Medicine, Division of Gastroenterology and Hepatology, Chung Shan Medical University Hospital, Taichung 402, Taiwan; chen.wl0204@gmail.com; 6Department of Orthopaedic Surgery, National Taiwan University Hospital, Taipei 100, Taiwan; wukuanwen@gmail.com; 7Department of Internal Medicine, Division of Endocrinology and Metabolism, Chung Shan Medical University Hospital, Taichung 402, Taiwan

**Keywords:** denosumab, glucolipotoxicity, pancreatic β-cell, RANK/RANKL pathway, type 2 diabetes

## Abstract

Obesity is strongly associated with insulin sensitivity in type 2 diabetes (T2D), mainly because free fatty acids (FFAs) are released from excess fat tissue. Long-term exposure to high levels of FFAs and glucose leads to glucolipotoxicity, causing damage to pancreatic β-cells, thus accelerating the progression of T2D. Therefore, the prevention of β-cell dysfunction and apoptosis is essential to prevent the development of T2D. Unfortunately, there are currently no specific clinical strategies for protecting β-cells, highlighting the need for effective therapies or preventive approaches to improve the survival of β-cells in T2D. Interestingly, recent studies have shown that the monoclonal antibody denosumab (DMB), used in osteoporosis, displays a positive effect on blood glucose regulation in patients with T2D. DMB acts as an osteoprotegerin (OPG) by inhibiting the receptor activator of the NF-κB ligand (RANKL), preventing the maturation and function of osteoclasts. However, the exact mechanism by which the RANK/RANKL signal affects glucose homeostasis has not been fully explained. The present study used human 1.4 × 10^7^ β-cells to simulate the T2D metabolic condition of high glucose and free fatty acids (FFAs), and it investigated the ability of DMB to protect β-cells from glucolipotoxicity. Our results show that DMB effectively attenuated the cell dysfunction and apoptosis caused by high glucose and FFAs in β-cells. This may be caused by blocking the RANK/RANKL pathway that reduced mammalian sterile 20-like kinase 1 (MST1) activation and indirectly increased pancreatic and duodenal homeobox 1 (PDX-1) expression. Furthermore, the increase in inflammatory cytokines and ROS caused by the RANK/RANKL signal also played an important role in glucolipotoxicity-induced cytotoxicity, and DMB can also protect β-cells by reducing the mechanisms mentioned above. These findings provide detailed molecular mechanisms for the future development of DMB as a potential protective agent of β-cells.

## 1. Introduction

It is commonly accepted that obesity is highly associated with insulin insensitivity in patients with type 2 diabetes (T2D). Specifically, excess body adiposity, especially visceral fat, continuously releases free fatty acids (FFAs) into the circulation, which has been implicated in the pathogenesis of obesity-related insulin resistance in peripheral tissues [1]. This leads to the underuse of blood glucose and explains the underlying mechanism by which being overweight or obese increases the chance of developing T2D. In fact, long-term exposure to high levels of FFAs and glucose results in damage to various tissues of the body, a condition often referred to as glucolipotoxicity, particularly harmful to pancreatic β-cells [2]. Furthermore, hyperglycemia also increases the workload of β-cells, and it exacerbates glucolipotoxicity-induced β-cell exhaustion. This result, also known as β-cell failure, is the key stage that accelerates the progression of T2D in the midterm [3]. Therefore, the prevention of β-cell dysfunction and glucolipotoxicity-caused apoptosis is essential to block the progression of the disease. Unfortunately, no specific clinical strategy has been developed to protect β-cells to date. This indicates that effective treatments or prevention strategies are urgently needed to improve the survival of T2D β-cells.

Interestingly, some recent studies have shown that denosumab (DMB), a recombinant monoclonal human osteoporosis antibody used to treat osteoporosis, has positive effects on glucose homeostasis in patients with T2D [4]. DMB is a mimetic agent similar to osteoprotegerin (OPG) that inhibits osteoclast maturation, function, and survival by blocking the action of the receptor activator of the NF-κB ligand (RANKL) [5]. It is known that bone-forming osteoblasts and bone-resorption osteoblasts interact to regulate bone remodeling. Therefore, DMB is believed to play a biological role by suppressing RANKL signals [6]. Increasing evidence has confirmed the relationship between diabetes and osteoporosis. Compared to the general population, patients with diabetes are more likely to have osteoporosis. The pathological mechanisms are not yet conclusive, and the reason is speculated to be a negative balance of bone remodeling. Therefore, DMB is one of the common therapeutic agents for patients with diabetic osteoporosis. [7]. Interestingly, recent research has shown that DMB appears to have the ability to improve the homeostasis of blood sugar levels. In patients with diabetes, there is evidence that fasting serum glucose levels are slightly reduced with DMB, suggesting that blocking RANKL may have a clinically important effect on glucose metabolism [8]. However, how exactly RANK/RANKL signals affect glucose homeostasis has not yet been determined.

Among the various mechanisms involved in the regulation of the blood glucose balance of RANKL signals, pancreatic cells are considered to be one of the possible targets. In fact, excessive RANKL signals have been identified as an important contributor to β-cell dysfunction and obesity-related T2D [9]. Furthermore, both DMB and OPG stimulate cell proliferation. This increased the mass of β-cells in diabetic mice and significantly attenuated hyperglycemia, suggesting the potential for the use of osteoporosis drugs for the treatment of T2D [10]. In particular, the transcriptional factor pancreatic and duodenal homeobox 1 (PDX-1) is known to be the main regulator of β-cells’ function and survival, especially in the pathological state of T2D [11]. Diabetes-related glucolipotoxicity is known to lead to the increased activity of mammalian sterile 20-like kinase 1 (MST1), which directly phosphorylates PDX-1 and degrades it through the proteasomal pathway [12]. As a result, maintaining PDX-1 levels via pharmacological activation is considered a possible effective strategy to alleviate T2D-related β-cell exhaustion. Furthermore, inflammation and the accumulation of reactive oxygen species (ROS) in the pancreatic islet are also considered important causes of T2D-related cell dysfunction [13]. Other studies have shown that ROS accumulation is strongly related to MST1 activation [14], and we have also found that oxidative stress is an important regulator of β-cell apoptosis and dysregulation in glucolipotoxicity [15]. Since RANK/RANKL can play a role in the cell by increasing ROS accumulation [16], it is reasonable to speculate that DMB may reduce MST1 activation via this pathway. However, it is still unclear whether DMB has a protective effect on β-cells through these mechanisms. Therefore, in the present study, we used the human β-cell line 1.4 × 10^7^ to simulate T2D under high-glucose and high-FFA conditions, investigated the ability of DMB to protect β-cells from glucolipotoxicity, and determined the possible molecular mechanisms of its effects.

## 2. Results

### 2.1. High Glucose and High FFA Levels Significantly Induce Glucolipotoxicity in 1.4 × 10^7^ Human Pancreatic β-Cells

First, we established a culture environment with high glucose and high FFA levels, similar to that found in diabetes in the human β-cell line 1.4 × 10^7^. This is an insulin-producing pancreatic islet cell line and is therefore considered suitable for research on diabetes [17]. As shown in Figure 1a, with an increasing concentration of FFAs, the survival rate of the β-cells gradually decreased and reached about half the viability of 250 μM for 24 h. At the same time, increasing the glucose concentration to 10.0 g/L exacerbated the survival rate, indicating that the cells are actually affected by glucolipotoxicity. Morphological observations also obtained similar results demonstrating that the β-cells showed obvious cytotoxicity after continuous treatment with high glucose (HG, 10.0 g/L) and high FFA levels (250 μM) for 24 h (Figure 1b). Thus, in subsequent experiments, the cells were treated under this condition as models of glucolipotoxicity. In order to analyze the type of cell death caused by HG + FFAs in the cells, we used an automatic fluorescent cell counter to analyze the results of the double fluorescent staining of acridine orange/propidium iodine (AO/PI). The results show that the percentage of apoptotic cells in high FFAs increased significantly. At this point, when HG was added simultaneously, the level of apoptosis could be further increased (Figure 1c). The above results demonstrate that HG and high FFAs can actually cause the glucolipotoxicity of 1.4 × 10^7^ human pancreatic β-cells and lead to cellular apoptosis, consistent with the current clinical understanding of β-cell exhaustion associated with diabetes [18].

### 2.2. DMB Attenuates Glucolipotoxicity-Induced Apoptosis of 1.4 × 10^7^ β-Cells

Next, we tested whether DMB could reduce the damage caused by glucolipotoxicity in β-cells. The MTT results showed that 24 h of cotreatment with 0.1 μM DMB provided optimal protection against cell damage by high glucose and FFA levels, with a significant increase in survival from 47% to 62% (Figure 2a). Therefore, in subsequent experiments, the concentration of DMB was treated with 0.1 μM. Afterwards, we used terminal deoxynucleotidyl transferase dUTP Nick-End Labeling (TUNEL) assays to detect apoptosis. As shown in Figure 2b, about 28% of β-cells showed TUNEL-positive signals. In contrast, the TUNEL-positive cell ratio after cotreatment with 0.1 μM DMB was observed to be significantly reduced to 12%, demonstrating that DMB can actually reduce the apoptosis caused by HG and high FFAs. Western blots further confirmed that two well-known typical apoptosis markers, cleaved caspase3 and PARP, increased significantly after 24 h of HG and high FFA treatment. On the contrary, while treating 0.1 μM DMB simultaneously, these apoptotic markers decreased significantly (Figure 2c). These results indicate that DMB actually mitigated the damage caused by high glucose and high FFA levels and that it reduced the rate of apoptosis in the 1.4 × 10^7^ β-cells.

### 2.3. DMB Reduces MST1 Activation by Suppressing RANKL/RANK Signals in 1.4 × 10^7^ β-Cells

We previously reported that β-cells activate MST1 and reduce the expression of PDX-1 in response to glucolipotoxicity [15]. Therefore, maintaining PDX-1 expression in cells is essential to increase survival under stress conditions. As a result, we detected changes in PDX-1 and MST1 in the β-cells. The results of the Western blotting in Figure 3a show that high glucose and FFA levels significantly increased the expression of cleaved MST1 and reduced the level of PDX-1 in the β-cells. In contrast, cotreatment with DMB reversed the situation found above by significantly reducing MST1 cleavage and increasing PDX-1 expression, suggesting that DMB may protect β-cells from glucolipotoxicity by regulating the signals of MST1/PDX-1. Previous reports have shown that MST1 activation is related to the pro-inflammatory signaling pathways activated by the nuclear factor NF-κB. As a result, we measured the activation of NF-κB signals by detecting p65 phosphorylation in Ser^536^. Figure 3a reveals that phospho-NF-κB p65 tends to increase by 12 times in the presence of glucolipotoxicity, while DMB effectively reduces the degree of p65 phosphorylation. The results of qPCR in Figure 3b further confirm that pro-inflammatory cytokines induced by high glucose and FFA levels, including IL-1β, IL-6, and TNFα, can be significantly inhibited by 24 h treatment with DMB. This suggests that DMB may actually reduce the inflammation and MST1 activation caused by β-cell glucolipotoxicity. Since NF-κB signals are activated by the RANK/RANKL pathway, we then investigated the causal relationship between NF-κB and MST1. In the case of glucolipotoxicity, p65 phosphorylation was significantly reduced in RANK knockdown; however, MST1 knockdown did not affect p65 phosphorylation (Figure 3c). These results indicate that DMB may reduce MST1 activation by suppressing RANK signals. The results of the MTT also found that the protective effect of DMB on glucolipotoxicity in β-cells can be significantly restored by RANK or MST1 knockdown, suggesting that RANK signals can play an important role in the above mechanisms (Figure 3d).

### 2.4. DMB Protects β-Cells against Glucolipotoxicity by Preserving Mitochondrial Function and Insulin Production/Secretion

Previous studies have demonstrated that glucolipotoxicity also activates MST1 by increasing oxidative stress damage and therefore leads to β-cell dysfunction. To confirm that DMB can also alleviate the oxidative stress induced by high glucose and FFA levels, we analyzed the membrane potential state of mitochondria, an imbalance of which leads to excessive ROS production by β-cells. As shown in Figure 4a, the results of JC-1 staining showed that HG + FFAs reduced mitochondrial membrane potential (green fluorescent signals) and indicated that mitochondrial functions may have been damaged. On the contrary, DMB treatment effectively restored the mitochondrial membrane potential (red fluorescent signals). The staining of ROS probe 2′, 7′dichlorofluorescin diacetate (DCFH-DA) also revealed that DMB could effectively reduce ROS accumulation in β-cells, in accordance with the results of the mitochondrial membrane potential (Figure 4b). It is now known that excessive damage to oxidative stress eventually leads to a decrease in β-cell functions, thus blocking insulin production and secretion. Therefore, we measured insulin concentrations in a cell culture medium using ELISA (Invitrogen, Carlsbad, CA, USA). As shown in Figure 4c, DMB treatment can significantly restore HG + FFA-inhibited insulin secretion. The qPCR results also illustrated that DMB significantly increased the expression of glucolipotoxicity-inhibited proinsulin mRNA, indicating that it actually effectively restored insulin production efficiency. Interestingly, although RANK mRNA expression did not appear to have changed, RANKL mRNA expression was significantly induced by glucolipotoxicity. However, DMB effectively inhibited the up-regulation of RANKL mRNA expression, which means that blocking RANK/RANKL signals may explain how DMB protects β-cells under HG + FFA (Figure 4d). Finally, we also investigated the changes in the expressions of various proteins involved in the reduction of oxidative stress. As shown in Figure 4e, the immunoblotting results showed that high glucose and FFA levels significantly inhibit the expressions of the main free radical scavenging proteins, including catalase and superoxide dismutase type 1 and type 2 (SOD1/2). In contrast, cotreatment with DMB at the same time could restore the expressions of these scavenger proteins. It is interesting to note that, when the expression of sirtuin 1 (Sirt1) was suppressed, DMB could be inhibited from restoring the expression of antioxidant proteins and the peroxisome proliferator-activated receptor-gamma coactivator-1α (PGC-1α) level, which is essential for maintaining mitochondrial function. This suggests that the up-regulation of Sirt1 may also be another mechanism by which DMB can play a protective role in β-cells.

## 3. Discussion

It is estimated that more than half of patients with T2D eventually need insulin therapy due to pancreatic cell dysfunction [19]. In β-cells, it is generally believed that high-glucose- and high-FFA-induced glucolipotoxicity is the most important cause of apoptosis and the failure of function. However, there are still no effective β-cell protective drugs in clinical practice. As a result, in our current study, we found that DMB appears to be able to effectively reduce damage to the glucolipotoxicity of β-cells, thus restoring normal cell functions, including insulin production and secretion. At the same time, DMB can increase PDX-1 expression in β-cells under glucolipotoxicity, which explains why DMB contributes to maintaining blood glucose homeostasis from the molecular mechanism. In the mature pancreas, only β-cells are recognized as capable of expressing PDX-1. Its main function is known to promote insulin production and synthesis, and it is closely related to maintaining the normal physiological function of β-cells. Evidence has indicated that glucolipotoxicity can cause β-cell dysfunction by inhibiting PDX-1. On the contrary, once PDX-1 expression is maintained, it can effectively protect cells to maintain insulin production and secretion while reducing dysfunction and apoptosis. In this study, we found that DMB increased PDX-1 expression by inhibiting MST1 activation. In particular, we also revealed that MST1 can be activated by RANK/RANKL signals, indicating that DMB can reduce MST1 activation by blocking the RANK pathway and indirectly increase the expression of PDX-1. In fact, many studies show that patients with T2D have a relatively high expression of RANKL [20]. It has also been confirmed that a large amount of RANKL is closely associated with systemic insulin resistance and may be caused by RANK-related inflammatory responses [21]. In our current study, our results also prove that high glucose and FFA levels actually promote RANKL mRNA expression in β-cells, supporting the inferences of previous related studies.

Furthermore, mitochondrial dysfunction and oxidative stress are considered to be important causes of β-cell failure associated with T2D, particularly in environments of high glucose and FFA levels [22]. In cells, long-term excessive glucose and FFA levels change β-cell metabolism and cause defects in mitochondrial function. Damaged mitochondria lead to excessive ROS accumulation in cells, which inhibits the efficiency of glucose-induced insulin production and secretion. Excess oxidative stress can cause many negative effects in the cell, one of which is associated with the degradation of PDX-1. Oxidative stress can activate intracellular caspases, which cleaves MST1 from the proenzyme state to the active state. Activated MST1 is directly phosphorylated at Thr^11^ of PDX-1, inducing PDX-1 to translocate from the cytoplasm to the nucleus and enter the degradation pathway of the ubiquitin proteasome system (UPS) [12]. Furthermore, MST1 has been found to be associated with promoting cell senescence [23]. Overactive MST1 suppresses intracellular antioxidant genes and makes cells more vulnerable to stress conditions [24]. Similarly, previous studies have shown that MST1 exacerbates cell senescence by inhibiting a protein called Sirt1 [25]. Sirt1 has long been considered to be an anti-aging protein that can reduce the damage caused by oxidative stress and suppress cell aging by influencing a group of downstream antioxidant genes called vitagenes, including catalases and superoxide dismutase (SOD) [26]. Furthermore, Sirt1 can also improve mitochondrial quality and biogenesis by activating Nrf2 and PGC-1α, thereby improving antioxidant scavenging capacity [27]. Parallelly, our results show that DMB can restore the expression of Sirt1, which is inhibited by glucolipotoxicity, thus increasing the expressions of SOD1/2, catalase, and PGC-1α. This may also provide an explanation for other mechanisms that suggest the possible role of DMB in the independent pathway of RANK signals. However, other studies have also shown that increased Sirt1 has positive effects on the reduction in NF-κB-p65 and IL-1 inflammation in pancreatic tissues and can also help improve pancreatic activity [28]. All of the results above suggest that the protective effect of DMB on β-cells may be multivariable and may not be limited to the RANK/RANKL pathway, but detailed mechanisms still need to be investigated for clarification.

Recent clinical data have indicated that DMB significantly improves glycemic parameters and shows a greater effect in patients with impaired glucose tolerance (IGT) [29]. By definition, IGT occurs when the blood glucose level is higher than normal, although the patient is usually symptomless at that time, but it can already be defined as prediabetes. However, it is now known that β-cell function is impaired during this period and that cell numbers have already declined significantly [30]. These facts show that protection specifically for β-cells needs to be carried out as soon as possible, but in clinical trials, there is currently no effective drug. Kondegowda et al. previously found that DMB inhibits the interaction of RANK/RANKL and stimulates β-cell replication, but they did not explain the mechanism in detail [10]. In addition to supporting many of the results mentioned above, our current findings provide a more detailed mechanical explanation of this basis, which will be beneficial for the future development of DMB as a potential protective agent for β-cells. However, despite our results showing that the use of DMB alone does not seem to have a significant adverse effect on 1.4 × 10^7^ cells, the exact situation in future animal studies should be confirmed to verify its utility and safety considerations. In conclusion, our findings demonstrate that DMB effectively reduced cellular dysfunction and apoptosis caused by high glucose and FFA levels in 1.4 × 10^7^ human pancreatic β-cells. In addition, the inflammatory cytokines and ROS elevation caused by RANK/RANKL signals also play an important role in glucose-induced cytotoxicity, and DMB protects β-cells by attenuating the above-mentioned mechanism.

## 4. Materials and Methods

### 4.1. Materials

All common chemicals, including methylthiazol-2-yl-3 (4,5-methylthiazol-2-yl)-2,5-diphenyltetrazolium bromide (MTT), acridine orange (AO), propidium iodine (PI), 2,7-dichlorodihydrofluorescein diacetate (DCFH-DA), and JC-1, were purchased from Sigma (München, Germany). Antibodies against caspase 3, poly(ADP-ribose) polymerase (PARP), PDX-1, MST1, catalase, Nrf2, and PGC-1α were obtained from Santa Cruz Biotechnology (Santa Cruz, CA, USA); β-actin was from Novus Biotechnology (Littleton, CO, USA); RANK was from Cell Signaling Technology (Danvers, MA, USA); pSer536-p65 was from Abcam (Cambridge, MA, USA); and SOD1, SOD2, and Sirt1 antibodies were from GeneTex (Irvine, CA, USA). Denosumab was purchased from GlaxoSmithKline (London, UK). Lentiviruses carrying MST1, RANK, and Sirt1 specific short hairpin (sh)RNAs were purchased from the Taiwan National RNAi Core Facility. All chemicals were dissolved in phosphate-buffered salt (PBS) solution and stored at 20 °C until use in experiments.

### 4.2. Cell Culture and Viability Assay

Human 1.4 × 10^7^ pancreatic β-cells were purchased from the European Collection of Authentic Cell Cultures (ECACC) in London, UK. The cells were grown with RPMI-1640, supplemented with 10% fetal calf serum, antibiotics (100 mg/mL penicillin, 100 mg/mL streptomycin), and 2 mg/mL L-glutamine, and they were maintained in humid air containing 5% CO_2_ at 37 °C. To test viability, the cells were treated with MTT tetrazolium salt for 30 min and analyzed spectrophotometrically at 550 nm. Viability was determined by the percentage of control cells treated with the vehicle alone. The average population of the control cells was set to 100% to compare the survival rate of the other cells tested.

### 4.3. Acridine Orange (AO)/Propidium Iodide (PI) Assay

To accurately determine cell survival, we used an AO/PI staining kit (Logos Biosystems, Annandale, VA, USA). In short, AO penetrates the living cells and the dead cells and generates green fluorescence. However, PI only enters dead cells and paints all dead nucleated cells to generate red fluorescence. Once the nucleus is surrounded, the fluorescence of PI increases by 20 to 30 times, causing the cell to glow red. For AO/PI, the cells were trypsinized and suspended after treatment according to the specified conditions. Subsequently, the 2 μL dye solution was mixed with 18 μL cell samples, and, following the manufacturer’s protocol, we directly added cell measurements and analyses of the viability of the Luna-FL automated double fluorescent cell counter (Logos Biosystems) to the cell sample. The PI-stained apoptotic cells were quantified by comparing the cell count of five independent samples. Values are expressed in terms of the percentage of dead cells compared to the total number of cells.

### 4.4. TUNEL Assays

The TUNEL assays were performed according to the manufacturer’s instructions (Thermo Fisher Scientific, Waltham, MA, USA). In short, the cells were fixed at 37 °C for 15 min with 4% paraformaldehyde. Subsequently, the cells were permeabilized for 15 min at 37 °C with 0.2% Triton X-100, washed with PBS, and incubated for 10 min at 37 °C with terminal deoxynucleotidyl transferase buffer (TdT). After incubation, the cells were mixed with the TdT reaction mixture and incubated for 1 hr at 37 °C. The cells were then washed with 3% BSA, and fluorescent dyes were added at 37 °C for 30 min. After washing the PBS, the final concentration of H33258 was 2 g/mL for the visualization of the nucleus, and the break of the DNA strand was visualized with a fluorescence microscope (DP72/CKX41, Olympus, Tokyo, Japan). The percentage of TUNEL-positive cells was calculated in five randomly selected areas of each group.

### 4.5. Western Blot Analysis

After treatments, the cells were harvested and homogenized using Gold lysis buffers (50 mM Tris-HCl, pH 8.0, 5 mM ethylenediaminetetraacetic acid, 150 mM NaCl, 0.5% nonidet P-40, 0.5 mM phenylmethylsulfonyl fluoride, and 0.5 mM Dithiothreitol) to extract entire cells. After the protein concentrations in each sample were measured, equal amounts (50 μg) of total proteins from cell lysate were separated from total cell lysate via electrophoresis with sodium dodecyl sulfate (SDS)–polyacrylamide gel and then transferred to a polyvinylidene difluoride (PVDF) membrane (Millipore). After blocking, the membranes were sequentially tested with primary antibodies and then with secondary antibodies combined with horseradish peroxidase. The primary antibodies were used diluted 1:1000 and secondary antibodies diluted 1:5000 in 0.1% Tween-20. Then, using Amersham ECL detection agents, proteins on the PVDF membrane were demonstrated, and images were obtained using an AI600 imaging system (GE Healthcare, Chicago, IL, USA). The relative protein expression levels were densitometrically quantified using ImagePro Plus 6.0 software (Media Cybernetics, Silver Spring, MD, USA), standardized based on the protein expression levels of β-actin, and compared to the normalized protein levels of the control cells. The control protein level was set to 1.0 for comparison, and the results of these calculations are representative of three independent experiments.

### 4.6. mRNA Expression Analysis Using Reverse-Transcription Quantitative PCR (qPCR)

Total RNA was extracted from the cells using an RNeasy kit following the manufacturer’s protocol (Qiagen, Hilden, Germany). After the RNA was purified, following the manufacturer’s recommendation, we changed the mRNA to cDNA using the TProfessional Thermocycler (Biometra, Göttingen, Germany). Then, qPCR was performed using the Power SYBR Green PCR Master Mix (Applied Biosystems, Foster City, CA, USA) on an ABI 7300 sequence detection system (Applied Biosystems, Foster City, CA, USA). The reverse transcription process was performed with the following temperature parameters: initial denaturation at 95 °C for 10 min; 40 cycles of denaturation at 95 °C for 15 s; annealing at 60 °C for 1 min; and dissolution phase at 95 °C for 15 s, 60 °C for 15 s, and 95 °C for 15 s. The following primer pairs were used: forward 5′-CACCT CTCAA GCAGA GCACA G-3′ and reverse 5′-GGGTT CCATG GTGAA GTCAA C-3′ for IL-1β; forward 5′-AGGGC TCTTC GGCAA ATGTA-3′ and reverse 5′-GAAGG AATGC CCATT AACAA CAA-3′ for IL-6; forward 5′-AAATG GGCTC CCTCT CATCA GTTC-3′ and reverse 5′-TCTGC TTGGT GGTTT GCTAC GAC-3′ for TNFα; forward 5′-ACACC TGTGC GGCTC ACA-3′ and reverse 5′-TCCCG GCGGG TCTTG-3′ for proinsulin; forward 5′-TGGCC CGGAT GAATA CYYGG-3′ and reverse 5′-GCACA CTGTG TCCTT GTTGA G-3′ for RANK; forward 5′-ATTGT CCAGT CGCAC TTCGT-3′ and reverse 5′-AGTCG AGTCC TGCAA ACCTG-3′ for RANKL; forward 5′-TGGTAT CGTGG AAGGA CTCAT GAC-3′ and reverse 5′-ATGCC AGTGA GCTTC CCGTT CAGC-3′ for glyceraldehyde-3phosphate dehydrogenase (GAPDH). The mRNA expression levels of all target genes were normalized to GAPDH expression, and the cDNA sample was measured in three independent experiments. The relative expression values of mRNA were obtained using sequence detection system software with delta-delta Ct (Sequence Detection System v1.2.3-7300 real-time PCR system; Applied Biosystems).

### 4.7. Analysis of the Mitochondrial Membrane Potential

To evaluate the potential of mitochondrial membranes, the cells were incubated at 37 °C in a fresh medium containing 1 µM JC-1 for 30 min. At the end of the incubation, the cells were washed and photographed with a reverse fluorescent microscope (DP72/CKX41, Olympus). Image Pro Plus 6.0 software (Media Cybernetics, Rockville, MD, USA) was used to measure red/green fluorescence, and the results show an average red/green fluorescence intensity ratio. All results were calculated via a statistical analysis using five random images without adjacent locations in each group.

### 4.8. Measurement of Reactive Oxygen Species (ROS)

To measure the ROS levels within the cell, 20 μM DCFH-DA was incubated in 0.5 h at 37 °C at 5% CO_2_. After incubation, the cells were harvested and immediately rinsed twice with PBS to remove background signals. To quantify the intracellular ROS content, we used multidetection readers at 485 and 535 nm excitation and emission wavelengths (SpectraMax iD5 microplate readers, Molecules, Sunnyvale, CA, USA). The relative fluorescence intensity was considered to be the average of three repeated experiments. Fluorescence images were also collected using optical microscopes (DP72/CKX41), all of which used the same fluorescent conditions and exposure time.

### 4.9. ELISA Measurements of TNFα, IL-6, and IL-1β Content

The cells were placed overnight in 24-well plates with a density of 5 × 10^4^ cells/well and treated under the indicated conditions, followed by ELISA kits to quantify insulin concentrations in culture media according to the manufacturer’s instructions (Invitrogen, Carlsbad, CA, USA).

### 4.10. Statistical Analysis

All data are presented as average standard mean error (±SEM). Data were statistically analyzed using a variance analysis, followed by Dunnett’s multiple comparison post hoc test using SPSS v25.0 statistical software (SPSS, Inc., Chicago, IL, USA). The difference was considered statistically significant with values * for *p* < 0.05 and ** for *p* < 0.01.

## Figures and Tables

**Figure 1 ijms-24-10289-f001:**
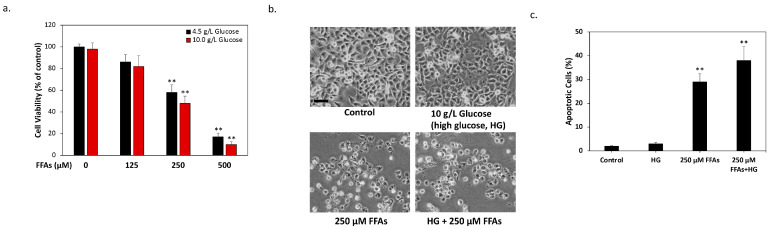
High levels of glucose and FFAs are associated with significant glucolipotoxicity in 1.4 × 10^7^ human pancreatic β-cells. (**a**) The results of the MTT assays showed a 42% cell death rate in the high FFA (250 μM) group and a 52% cell death rate in the high glucose (HG) + FFA group after 24 h of treatment compared to the control group. (**b**) Phase-contrast cell microscopic images were taken 24 h after treatment. Treatment with high glucose levels in 1.4 × 10^7^ β-cells caused morphological changes and cytotoxicity, while treatment with high glucose + FFAs further significantly reduced cell viability. The scale bar is 20 μm. (**c**) The results of the acridine orange/propidium iodide (AO/PI) staining demonstrated that, after 24 h of high glucose + FFA treatment, the rate of apoptotic cells increased significantly. All data were collected from at least three independent experiments and are presented as mean ± SEM. Significant differences were determined in multiple comparisons with Dunnett’s post hoc tests (** *p* < 0.01) compared to the control groups.

**Figure 2 ijms-24-10289-f002:**
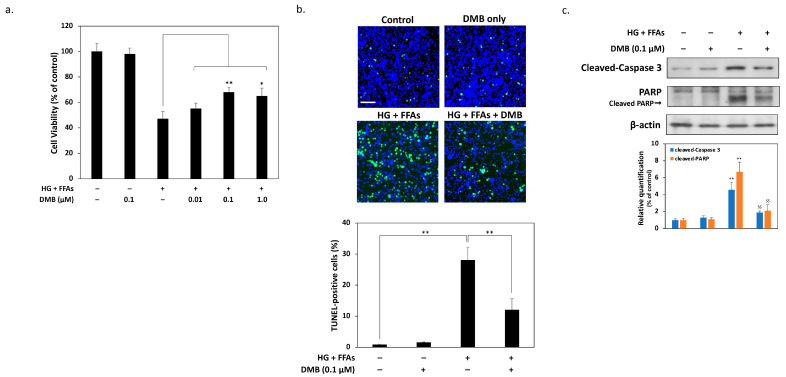
DMB reduces glucolipotoxicity-induced apoptosis in 1.4 × 10^7^ β-cells. (**a**) Cell viability was determined using MTT assays. The results show that 24 h of combined treatment with 0.1 μM DMB provided significant protection against cell death caused by high glucose and FFA levels. (**b**) TUNEL assays revealed that 28% of β-cells showed TUNEL-positive signals after 24 h of high glucose and high FFA treatment. On the contrary, the TUNEL-positive cell ratio was significantly reduced to 12% after treatment with 0.1 μM DMB. The scale bar is 200 μm. (**c**) Western blot analysis demonstrated that DMB inhibited caspase 3 and PARP activation during glucolipotoxicity, and the results of quantitative Western blots were also presented by densitometry. All data were collected from at least three independent experiments and are presented as mean ± SEM. Significant differences were determined in multiple comparisons with Dunnett’s post hoc tests compared to control groups (* *p* < 0.05 and ** *p* < 0.01) or compared to HG + FFA groups (^$$^
*p* < 0.01).

**Figure 3 ijms-24-10289-f003:**
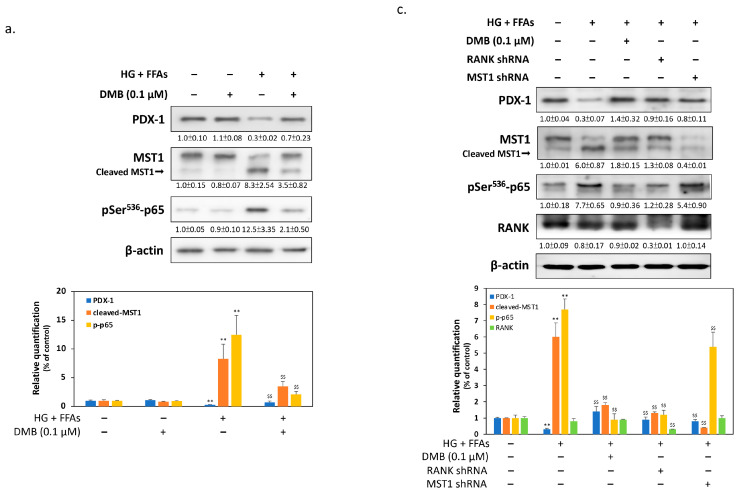
DMB reduces MST1 activation by suppressing RANK/RANKL signals in 1.4 × 10^7^ cells. (**a**) Immunoblotting results showed that PDX-1 expression markedly reduced and that the cleaved level of MST1 clearly increased when cells were treated with high glucose and FFA levels for 24 h. However, cotreatment with DMB reversed glucolipotoxic effects on the expressions of these proteins. (**b**) The results of qPCR demonstrated that, under glucolipotoxicity, DMB significantly reduced the expression levels of IL-1β, IL-6, and TNFα pro-inflammatory cytokines. (**c**) High glucose and FFA levels caused a marked reduction in PDX1. However, the knockdown of MST1 by shRNA partially restored this inhibition. In addition, the knockdown of RANK restored PDX-1 expression and simultaneously reduced NF-κB p65 phosphorylation, indicating that DMB could reduce MST1 activation by suppressing RANK signals. (**d**) The results of MTT demonstrated that the protective effect of DMB against glucolipotoxicity can be reduced by RANK or MST1 knockdown. All data were collected from at least three independent experiments and are presented as mean ± SEM. Significant differences were determined in multiple comparisons with Dunnett’s post hoc tests compared to control groups (** *p* < 0.01) or compared to HG + FFA groups (^$^
*p* < 0.05 and ^$$^
*p* < 0.01).

**Figure 4 ijms-24-10289-f004:**
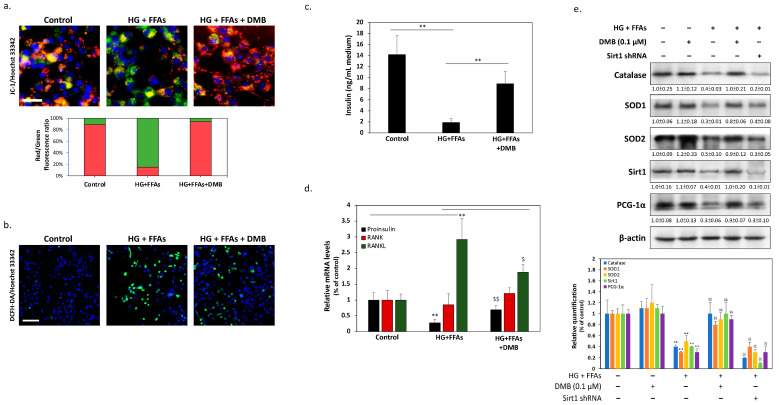
DMB protects β-cells from glucolipotoxicity by preserving mitochondrial functions and insulin production/secretion. (**a**) Immunofluorescent staining with JC-1. Green fluorescence indicated the dispersion of mitochondrial membrane potential after 24 h in β-cells treated with high glucose + FFA. Red fluorescence indicated that cotreatment with DMB effectively retained the potential of the mitochondrial membrane. The scale bar is 20 μm. (**b**) Intracellular oxidative burst was determined by 2′, 7′dichlorofluorescin diacetate (DCFH-DA) using fluorescent microscopy. The results show that DMB can effectively reduce ROS accumulation in β-cells induced by glucolipotoxicity. The scale bar is 200 μm. (**c**) Insulin concentrations in culture media were measured with a commercial ELISA kit. (**d**) Proinsulin, RANK, and RANKL mRNA levels were measured using qPCR. (**e**) The levels of some antioxidant signaling proteins, including catalases, SOD1, SOD2, Sirt1, and PCG-1α proteins, which maintain mitochondrial function, were analyzed using Western blotting. Furthermore, Sirt1 knockdown results showed that DMB effects on restoring antioxidant gene expression may be associated with the pathway of Sirt1. All data were collected from at least three independent experiments and are presented as mean ± SEM. Significant differences were determined in multiple comparisons with Dunnett’s post hoc tests compared to control groups (** *p* < 0.01) or HG + FFA groups (^$^
*p* < 0.05 and ^$$^
*p* < 0.01) or HG + FFAs + DMB groups (^§§^
*p* < 0.01).

## Data Availability

Data are contained within the article.

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
