# Peer review of "Denosumab Attenuates Glucolipotoxicity-Induced β-Cell Dysfunction and Apoptosis by Attenuating RANK/RANKL Signals"

_ijms, 2023, doi:10.3390/ijms241210289_

Round 1

Reviewer 1 Report

Lin et al reported that denosumb (DMB) could attenuate glucolipidptoxicity-induced beta-cells dysfunction and apoptosis through RANK/RANKL pathway. It has been reported that DMB and osteoprotegerin (OPG) can stimulate human beta cell proliferation (Kondegowda et al., 2015)., Cell Metabolism, 22, 77–85) T.he authors reported here that DMB increased cell viability from glucolipidotoxicity and reduced the rate of apoptosis. It will be interesting to see if this effect is due to the protection of 1.4e7 human pancreatic beta-cells, or due to the increase of cell proliferation. The increase in cell proliferation could also lead to what authors observed due to their compensation for the toxicity induced by glucose/free fatty acids. The authors should design an experiment to test the effects of DMB on cell proliferation. While the stimulation of proliferation could also be through RANK/RANKL pathway, it could help provide additional information.

For the Western blot, authors need to clarify if they have loaded the same amount of protein for each sample. Since the glucolipidptoxicity reduced cell viability, the observation they made may simply be due to having less protein overall. They should normalize (if they have not done so) by loading the same amount of protein and making comparisons. Also, for all the Western blot, how many replicates of the experiments? They should also do the quantitative analysis of the blot and the statistical analysis of the results. The same for qPCR, did they normalize the amount of mRNA from samples for qPCR testing?

English is fine.

Author Response

Reviewer#1

Lin et al reported that denosumb (DMB) could attenuate glucolipidptoxicity-induced beta-cells dysfunction and apoptosis through RANK/RANKL pathway. It has been reported that DMB and osteoprotegerin (OPG) can stimulate human beta cell proliferation (Kondegowda et al., 2015)., Cell Metabolism, 22, 77–85) T.he authors reported here that DMB increased cell viability from glucolipidotoxicity and reduced the rate of apoptosis. It will be interesting to see if this effect is due to the protection of 1.4e7 human pancreatic beta-cells, or due to the increase of cell proliferation. The increase in cell proliferation could also lead to what authors observed due to their compensation for the toxicity induced by glucose/free fatty acids. The authors should design an experiment to test the effects of DMB on cell proliferation. While the stimulation of proliferation could also be through RANK/RANKL pathway, it could help provide additional information.

Ans: Thanks for your valuable comments. Since this study is an in vitro experiment, it is difficult to clarify whether DMB has the ability to promote β-cell proliferation due to the fact that the cell line itself proliferates. However, at least from our results, it can be showed that the single use of DMB does not have an obvious impact on beta cells, including cell viability (Fig. 2). In the future, we have anticipated scheduling animal experiments on this issue and we expect to be able to better observe the effect of DMB (and safety) on beta cells in islets.

For the Western blot, authors need to clarify if they have loaded the same amount of protein for each sample. Since the glucolipidptoxicity reduced cell viability, the observation they made may simply be due to having less protein overall. They should normalize (if they have not done so) by loading the same amount of protein and making comparisons. Also, for all the Western blot, how many replicates of the experiments? They should also do the quantitative analysis of the blot and the statistical analysis of the results. The same for qPCR, did they normalize the amount of mRNA from samples for qPCR testing?

Ans: In order not to have any deviation, each well on SDS-PAGE was loaded with 50 ug total protein cell lysates during western blotting. Furthermore, in this revision we also added quantification to all WB blots and used β-actin as a normalization control, following the reviewer's comments. For qPCR, due to the requirements for primers and reaction solution, we used the ratio of 5λ mixture solution, 5λ cDNA sample and 10λ SYBR green master matrix to perform real-time PCR. All mRNA expression data will be normalized with GAPDH as the internal standard, and each cDNA sample was tested in three samples. These steps are carried out in accordance with the manufacturer's specifications (Applied Biosystems). We have also revised the above description in this revision, hoping to make the results of this study more clear and easy to read. Again, many thanks for your careful and deeply valuable comments.

Reviewer 2 Report

Type 2 diabetes (T2D) is the most common type of diabetes worldwide. Ranked as a global pandemic, it develops gradually, with physiological changes in the cells occurring before the onset of clinical symptoms. This work continues the study of the molecular mechanisms regulating β-cell apoptosis and glucolipotoxicity in these patient populations.

The authors showed that denosumab, a recombinant monoclonal human osteoporosis antibody, can reduce glucolipotoxic β-cell injury, restoring normal cell function, with insulin production and secretion.

This is a valuable manuscript on the important issue of T2D as well as in other clinical conditions. The manuscript is well written and the study well conducted. In my opinion, the manuscript is ready to be published.

Author Response

Reviewer#2

Type 2 diabetes (T2D) is the most common type of diabetes worldwide. Ranked as a global pandemic, it develops gradually, with physiological changes in the cells occurring before the onset of clinical symptoms. This work continues the study of the molecular mechanisms regulating β-cell apoptosis and glucolipotoxicity in these patient populations.

The authors showed that denosumab, a recombinant monoclonal human osteoporosis antibody, can reduce glucolipotoxic β-cell injury, restoring normal cell function, with insulin production and secretion.

This is a valuable manuscript on the important issue of T2D as well as in other clinical conditions. The manuscript is well written and the study well conducted. In my opinion, the manuscript is ready to be published.

Ans: Thank you very much for the suggestions of the reviewer. In this revision, we have modified some deficiencies in the text, including quantifying the results of all WB blots, strengthening the explanation of some experimental steps, and re-correcting the English writing of the full text. We would like to accept any further suggestions that could improve our work and hope that these modifications meet your standards and helps to improve the legibility and convincing of this study. Thanks again for your supportive comments.

Round 2

Reviewer 1 Report

While authors did the additional analysis on Westen Blot, they should clearly explain the number below the band (Fig. 3 and 4), I assumed they normalized the band intensity, with control as 1, but they need to clarify that. Further, I’d like to recommend authors make an additional bar graph to show the changes of each protein after the treatment, with statistical analysis, and describe the results in the result section.  

English is fine.

Author Response

Thanks for the reviewer's correction. In this version, we have quantified WB blots and drawn a bar graph for each protein in Fig. 3 and 4 (also marked with statistical symbols). Also, some minor bugs have been fixed. We hope these changes satisfy the reviewers, and we welcome any suggestions to improve our work.